# Molecular, Subcellular, and Arrhythmogenic Mechanisms in Genetic RyR2 Disease

**DOI:** 10.3390/biom12081030

**Published:** 2022-07-26

**Authors:** Ewan Douglas Fowler, Spyros Zissimopoulos

**Affiliations:** 1School of Biosciences, College of Biomedical and Life Sciences, Cardiff University, Cardiff CF10 3AX, UK; fowlered@cardiff.ac.uk; 2Institute of Life Science, Swansea University Medical School, Swansea University, Swansea SA2 8PP, UK

**Keywords:** ryanodine receptor, catecholaminergic polymorphic ventricular tachycardia, calcium release deficiency syndrome, delayed afterdepolarizations, early afterdepolarizations, calcium sparks, long QT syndrome, arrhythmias

## Abstract

The ryanodine receptor (RyR2) has a critical role in controlling Ca^2+^ release from the sarcoplasmic reticulum (SR) throughout the cardiac cycle. RyR2 protein has multiple functional domains with specific roles, and four of these RyR2 protomers are required to form the quaternary structure that comprises the functional channel. Numerous mutations in the gene encoding RyR2 protein have been identified and many are linked to a wide spectrum of arrhythmic heart disease. Gain of function mutations (GoF) result in a hyperactive channel that causes excessive spontaneous SR Ca^2+^ release. This is the predominant cause of the inherited syndrome catecholaminergic polymorphic ventricular tachycardia (CPVT). Recently, rare hypoactive loss of function (LoF) mutations have been identified that produce atypical effects on cardiac Ca^2+^ handling that has been termed calcium release deficiency syndrome (CRDS). Aberrant Ca^2+^ release resulting from both GoF and LoF mutations can result in arrhythmias through the Na^+^/Ca^2+^ exchange mechanism. This mini-review discusses recent findings regarding the role of RyR2 domains and endogenous regulators that influence RyR2 gating normally and with GoF/LoF mutations. The arrhythmogenic consequences of GoF/LoF mutations will then be discussed at the macromolecular and cellular level.

## 1. Introduction

Cardiac contraction is initiated by a wave of electrical activity (action potential) triggering a rapid rise in intracellular Ca^2+^ concentration, which activates sarcomere contraction, followed by repolarization and Ca^2+^ removal that allows the heart to relax and refill [1]. Gene mutations in any of the proteins involved in excitation–contraction coupling (ECC) can lead to arrhythmogenic cardiac disease, most commonly, long QT and Brugada syndromes (repolarization abnormalities), catecholaminergic polymorphic ventricular tachycardia (CPVT) (dysregulated intracellular Ca^2+^), and dilated and hypertrophic cardiomyopathies (sarcomere dysfunction) [2]. Cardiomyocyte Ca^2+^ homeostasis is indeed an arrhythmogenic substrate [3]. For contraction of the adult human heart, plasma membrane depolarization induces Ca^2+^ influx via voltage-gated Ca^2+^ channels (also known as L-type Ca^2+^ channel or dihydropyridine receptor) located on invaginations of the sarcolemma called transverse (T)-tubules. This by itself is insufficient to trigger sarcomere contraction, but the rapid rise in Ca^2+^ activates ryanodine receptors (RyR2) located on the immediately opposing junctional sarcoplasmic reticulum (SR) through which a much larger Ca^2+^ release occurs in the form of Ca^2+^ sparks [4]. The near-synchronous activation of ~10^4^ Ca^2+^ sparks at the start of the action potential produces the cytosolic Ca^2+^ transient that activates the contractile apparatus within cardiac myocytes. For relaxation, cytosolic Ca^2+^ concentration returns to baseline with Ca^2+^ extrusion outside of the cell via the Na^+^/Ca^2+^ exchanger (NCX) to remove an amount equal to the extracellular Ca^2+^ entry source, and Ca^2+^ uptake into the SR via the sarco/endoplasmic reticulum Ca^2+^ ATPase (SERCA) to remove an amount equal to the SR Ca^2+^ release source. Imbalances in cardiomyocyte Ca^2+^ cycling, namely the Ca^2+^ fluxes across the plasma and SR membrane, can alter the action potential and lead to arrhythmia.

The role of RyR2 in cardiac pathophysiology came to prominence more than 20 years ago. The first report implicating acquired RyR2 dysfunction in heart failure was published in 2000 [5], whereas a year later came the first studies associating mutations in the *RYR2* gene with catecholaminergic polymorphic ventricular tachycardia (CPVT) and arrhythmogenic right ventricular dysplasia type 2 (ARVD2) [6,7,8]. Recent advances in our knowledge of the molecular and arrhythmogenic mechanisms in genetic RyR2 disease are explored in detail in this mini-review.

## 2. RyR2 Structure-–Function Relationships

The RyR is the largest known ion channel, consisting of four identical subunits of ~5000 amino acids each, which combine to form a single Ca^2+^ permeable pore. Three RyR isoforms are expressed in mammals, RyR1 (the predominant isotype in skeletal muscle), RyR2 (the predominant isotype in cardiac muscle), and RyR3. Considering its size, complexity, and responsibility for achieving both short-(beat-to-beat) and long-term maintenance of heart contraction, it is not surprising that many factors fine-tune RyR2 function. In the following, we aim to describe determinants of RyR2 regulation as interacting layers consisting of: the pore-forming domains, domains that directly interact with the pore, domains that indirectly regulate the pore through intermediaries, or interacting regulatory proteins.

Recent high-resolution cryo-electron microscopy (EM) studies reveal that the long RyR polypeptide chain is organized in distinct structural domains that are predominantly α-helical and face the cytoplasm [9,10] (Figure 1). They include the N-terminus domain (NTD) composed of sub-domains A/B/C, three SPRY domains including RyR repeat (RY) 1&2, junctional solenoid (JSol), bridging solenoid (BSol) consisting of sub-domains B&C flanking RY3&4, and core solenoid (CSol). According to a different nomenclature by Peng et al. [10], JSol is known as the “handle domain”, BSol is known as the “helical domain” but it was previously referred to as the “central domain” [11], and CSol is also known as the “central domain”. To avoid confusion with “central domain” terminology, we adopt the nomenclature by des Georges et al. [9]. The C-terminal ~500 residues contain the transmembrane domain (TMD) composed of six segments that anchor the protein on the SR membrane, followed by the short cytosolic C-terminus domain (CTD). The pore is formed by the S6 lining helix supported by S5 and the S5-S6 loop containing the selectivity filter.

### RyR2 Channel Self-Assembly

Although RyR2 is a macromolecular complex with cytosolic, SR integral, and luminal accessory proteins (see Section 3.1), the 4-subunit RyR2 self-assembly is necessary and sufficient to form a fully functional Ca^2+^ release channel. To understand how RyR2 is auto-regulated, we consider a reductionist, “bottom-up” approach starting from the pore. Expression of the last two TM segments together with the CTD forms a tetrameric cation permeable pore that is constitutively open [12]. The CTD self-associates to support RyR tetramerization required for the formation of functional channels [13,14]. On top of the pore sits the CSol acting as the channels’ gatekeeper. The CSol makes extensive contacts with the pore-forming region including the S2S3 loop, the extended cytosolic part of S6, and the CTD [9,10]. However, the expression of large constructs encompassing the CSol, TMD, and CTD form ion channels that remain largely open and leaky [15,16,17]. Clearly, the remaining domains are needed to support the closure of the pore and fully regulate the channel. They are not making direct contact with the pore-forming region, but their effects are mediated via the CSol. Indeed, the NTD, Jsol, Bsol, and CSol are involved in multiple intra- and inter-subunit interactions with each other [9,10,18,19,20]. The NTD also interacts with itself to promote channel closure as well as to support the formation of functional tetrameric RyR2 channels [21,22,23]. RyR2 activity is fine-tuned by the peripheral SPRY and RY domains, and their interactions with JSol and BSol [9,10].

## 3. Physiological RyR2 Regulators

The principal physiological RyR2 regulator is Ca^2+^ acting from both the cytosolic and SR luminal side. RyR2 is mostly inactive at low diastolic (~100 nM) cytosolic Ca^2+^ concentration ([Ca^2+^]). Membrane depolarization causes a rapid rise in [Ca^2+^] from nanomolar to micromolar levels in the narrow dyadic cleft formed between T-tubule and SR junctional membrane, which increases RyR2 activity in a sigmoidal function of cytosolic Ca^2+^ concentration. RyR2 can be fully activated at ~100 μM Ca^2+^ (reviewed in [24,25]). Recent cryo-EM and mutagenesis studies suggest that the cytosolic Ca^2+^-binding site is located at the CSol–CTD interface with residues E3848, E3922, and T4931 (coordinates for human RyR2) critical for Ca^2+^ interaction [9,26,27,28]. Interestingly, although the Ca^2+^-bound RyR structure reveals a contraction of the CSol that facilitates the dilation of the S6 bundle, the pore remains closed [9,26]. This could be due to the in vitro experimental conditions of cryo-EM. RyR2 is also regulated by SR luminal Ca^2+^ (reviewed in [24,25]). Although luminal Ca^2+^ can indirectly activate RyR2 by acting on the cytosolic Ca^2+^-binding site following passage through the pore, a separate site accessible only to luminal Ca^2+^ is likely to exist. Mutagenesis analysis has indicated that residue E4872 within S6 is necessary for luminal Ca^2+^ activation without affecting cytosolic Ca^2+^ modulation [29].

### 3.1. Regulation of RyR2 by Accessory Proteins

RyR2 is also regulated by accessory proteins and post-translational modifications (phosphorylation and oxidation) (reviewed in [25,30]). Here, we focus on three RyR2-associated proteins, FK506-binding protein (FKBP), calmodulin (CaM), and calsequestrin (CSQ2) because of their implication in genetic cardiac disease.

#### 3.1.1. FKBP12/12.6

In native cardiac tissue, both of the two FKBP homologues of 12 and 12.6 kDa are found associated with RyR2 depending on the animal species [31]. RyR2 affinity is much greater for FKBP12.6 than FKBP12; however, native RyR2 occupancy by FKBP12.6 is minimal due to the latter being expressed at very low levels [31,32,33]. Functional studies generally suggest that FKBP12.6 promotes RyR2 channel closure and prevents SR Ca^2+^ leak (reviewed in [34]). Cryo-EM RyR structures in complex with FKBP12/12.6 indicate that the binding site is composed of multiple determinants contributed by the SPRY1, SPRY3, NTD, and JSol domains [26,35,36]. FKBP12.6 stabilizes RyR2 in a closed state by inducing the relaxation of the CSol and strengthening the NTD-NTD inter-subunit interaction [26].

#### 3.1.2. Calmodulin (CaM)

CaM is a high-affinity, Ca^2+^-binding, dumbbell-shaped protein containing four EF-hand motifs, two of each on both N and C lobes. Both the Ca^2+^ free (apo-CaM) and Ca^2+^ bound forms of CaM interact with RyR2 to inhibit channel activity, although inhibition is more pronounced at high Ca^2+^ [37,38]. The cryo-EM RyR2 structures in complex with apo-CaM and Ca^2+^-CaM indicate that they bind to distinct sites [39]. The N-lobe of apo-CaM binds BSolB, whereas its C-lobe binds at the interface between JSol and CSol. Ca^2+^-CaM shifts position and interacts only with the CSol. Ca^2+^-CaM stabilizes RyR2 in a closed state by inducing conformational changes in the CSol to release the pulling force for the dilation of the S6 bundle. Notably, CaM mutations have been linked with arrhythmia and sudden cardiac death, with most cases diagnosed as long QT syndrome and CPVT (reviewed in [40,41]). Mutations in all three *CALM* genes, which produce identical CaM peptide sequences, have been reported. Most of the disease-associated CaM mutations studied to date seem to reduce CaM’s inhibitory action on RyR2.

#### 3.1.3. Calsequestrin-2 (CSQ2)

CSQ2 is a low-affinity, high-capacity Ca^2+^-binding protein of ~45 kDa located within the SR lumen, where it buffers the free Ca^2+^ concentration to ~1 mM. It is associated with RyR2 via the SR integral proteins triadin and junctin, as well as directly interacting with the S1S2 luminal loop of RyR2 [42,43]. It is generally believed that CSQ2 exerts an inhibitory effect on RyR2 Ca^2+^ release (reviewed in [44]). CSQ2 knockout mouse models present with stress-induced arrhythmia indicating that CSQ2 plays an important role in the cardiac function [45,46]. Indeed, *CASQ2* is the second most mutated gene in CPVT accounting for ~5% of cases [41]. Initially described as the recessive form of CPVT (CPVT2), autosomal dominant mutations have also been reported [47,48,49]. CPVT-linked CSQ2 mutations may result in impaired Ca^2+^ buffering, multimer formation and/or RyR2 regulation [41,44,50].

To summarize, we propose a 4-layer model for cytosolic regulation of the RyR2 pore (Figure 2). The first layer directly blanketing the pore consists of the CSol. The second layer consisting of NTD, Jsol, and BSol transmits stimulatory/inhibitory signals to the pore via the CSol. The third layer consists of the SPRY and RY domains acting on the second layer. The final fourth layer consists of cytoplasmic ligands, namely accessory proteins and protein kinases. While small ligands such as Ca^2+^ may have access to the pore-forming region, the large size of proteins precludes them from doing so. The effects of cytoplasmic proteins feed into layers 1/2/3 to alter their conformation and RyR2 inter-domain interactions. For example, FKBP12.6 binding (to SPRY1, SPRY3, NTD, and JSol domains) directly affects layers 2&3, which in turn transmit an allosteric signal to the pore via layer 1. The effect of phosphorylation (mediated by protein kinases) also feeds into layers 1/2/3. S2808 and S2814, whose phosphorylation is implicated in RyR2 physiology and disease are located within RY3&4. The luminal regulation of the RyR2 pore is simpler. The effects of luminal ligands can be direct to the pore (e.g., luminal Ca^2+^) or via the small RyR2 luminal loops.

## 4. Cardiac Disease Associated with RyR2 Mutations

RyR2 mutations transmitted in an autosomal dominant manner account for ~70% of CPVT cases (CPVT1) [41,50]. CPVT is characterized by episodes of potentially lethal bidirectional or polymorphic ventricular tachycardia triggered by emotional or physical stress. CPVT is manifested in the absence of structural abnormalities, while electrocardiogram (ECG) recordings are normal at rest. It is estimated to affect 1:10,000 mostly young, seemingly healthy individuals. According to European Society of Cardiology guidelines [51], the first-line therapy for CPVT is beta-blockers to suppress the catecholamine stimulation of the β-adrenergic pathway. It was very recently reported that non-selective β-blockers (nadolol, propranolol) were more efficient at reducing the risk of life-threatening arrhythmic events compared with selective β-blockers [52]. If beta-blocker therapy is ineffective, additional treatment with flecainide is recommended. Flecainide is a Na^+^ channel inhibitor that may also inhibit RyR2 Ca^2+^ release [53] (see also [54,55]). Left cardiac sympathetic denervation is considered for patients who continue to experience arrhythmia despite optimal medication [51]. The implantable cardioverter defibrillator can be considered the last line of defense for patients who have survived a cardiac arrest or those who do not respond to optimal medication. However, its use is debated due to questionable efficacy and inappropriate shocks that may have a pro-arrhythmic effect [50].

Mutations in the *RYR2* gene are also causative of ARVD2 [8,56]. The clinical presentation of ARVD2 is very similar to CPVT including stress-induced ventricular tachycardia, but it is also associated with progressive degeneration and fibro-fatty replacement of the right ventricle. In addition, RyR2 mutations have also been implicated in non-stress-induced cardiac disease including long QT syndrome [57,58], torsade de pointes [59,60], left ventricular noncompaction [61,62], dilated cardiomyopathy [63,64], and hypertrophic cardiomyopathy [65]. Very recently, a new disease termed RyR2 Ca^2+^ release deficiency syndrome (CRDS) was described, underlined by RyR2 loss of function (LoF) mutations [66,67]. CRDS is characterized by ventricular arrhythmias and sudden cardiac death but a negative exercise stress testing for CPVT. A novel classification for genetic RyR2-related cardiac disease was recently proposed as a typical and atypical CPVT [41]. Typical CPVT as described above is due to CSQ2 and gain of function (GoF) RyR2 mutations. Atypical CPVT covers non-stress as well as stress-induced disease other than bidirectional or polymorphic ventricular tachycardia, due to RyR2 LoF or mutations in other genes such as *CALM*.

## 5. Molecular Mechanisms of Genetic RyR2 Disease

To date, more than 350 mutations have been reported including splicing, deletions, insertions, and nonsense mutations, but the vast majority are missense (for an up-to-date list, see the Human Gene Mutation Database (HGMD^®^), http://www.hgmd.cf.ac.uk/ac/gene.php?gene=RYR2) (accessed on 25 July 2022). Pending further characterization, most of these mutations are currently classified as variants of unknown significance [68]. Mutations are found throughout the RyR2 peptide sequence but tend to concentrate on five structural domains, NTD (16.9%), BSolB (19.8%), CSol (21.6%), TMD (16%), and CTD (4.1%) (Figure 1 and Table 1). This is more evident once the amino acid length of each domain is taken into consideration. If we consider how many amino acid variants have been reported within each individual domain, NTD (9.1%), BSolB (11.9%), CSol (12.5%), TMD (13.7%), and CTD (17.5%), the small CTD is, in fact, the most frequently mutated structural element. In terms of disease diagnosis, there is no apparent correlation between the location of mutations on the RyR2 structure and the clinical phenotype presenting as typical or atypical CPVT. On the other hand, the few deletions, insertions, and nonsense mutations reported to date are often associated with atypical CPVT or cardiomyopathy irrespective of their location on the RyR2 structure (e.g., [61,63,67,69]).

Most pathological RyR2 missense mutations are GoF and causative of typical CPVT. They may show increased sensitivity to either the cytosolic [70,71] or luminal Ca^2+^ activation [72]. It has also been suggested that the GoF phenotype is due to reduced FKBP12.6 affinity (unmasked following RyR2 phosphorylation) [73] (see also [74,75]). Similarly, reduced CaM affinity has also been proposed to contribute to the GoF phenotype [76]. Moreover, altered RyR2 inter-domain interactions have been shown to underlie the GoF phenotype [19,20,22,77].

Based on their observations, three labs have put forward their own hypotheses to explain how RyR2 GoF mutations result in aberrant cardiomyocyte Ca^2+^ cycling. The first hypothesis by Marks and colleagues is that mutations impair the binding of FKBP12.6 which helps maintain the channel closed in the diastolic phase [73]. The second hypothesis by Yano and colleagues is that mutations impair the inter-domain interaction between the NTD and BSolB to destabilize the closed conformation of the channel [20]. The third hypothesis by Chen and colleagues is that mutations enhance the sensitivity to SR luminal Ca^2+^ thereby increasing the propensity for arrhythmogenic spontaneous Ca^2+^ waves [72]. However, these three hypotheses can be refined to include additional observations. The first hypothesis can be expanded to include other accessory proteins such as CaM [76]. The second hypothesis can be generalized to include altered RyR2 intra- and inter-subunit interactions, such as the JSol-BSolB interaction [19], NTD self-association [22], and NTD-CSol interaction [77]. The third hypothesis can be expanded to include mutations that enhance the sensitivity to cytosolic Ca^2+^ [70,71]. In addition, these three hypotheses are not mutually exclusive, with individual mutations potentially operating via different mechanisms. Moreover, the same mutation may affect more than one mechanism. For example, the R2474S is found to disrupt the NTD-BSolB interaction [20], which in turn impairs the RyR2 interaction with CaM [76].

Unlike GoF mutations, LoF mutations are associated with CRDS and other forms of atypical CPVT. Initially only a handful, >20 have been reported in the last couple of years as the focus has shifted to patients with atypical CPVT. Most of them are missense mutations, but for deletion and a large duplication/insertion, found within the CSol and TMD (Table 2). A common defect of LoF mutations is the perturbation of cytosolic and/or luminal Ca^2+^ activation [66,78]. Alternatively, they may affect the CSol-TMD interaction or the Ca^2+^-conducting properties of the pore. Other LoF mutations (e.g., those occurring within the CTD) may impair RyR2 tetramerization that is necessary for the formation of a functional channel, as reported for L433P (which has both GoF and LoF characteristics) [22].

Thus, mutations can be either GoF or LoF and result in typical or atypical CPVT, respectively, irrespective of their location on the RyR2 structure. We propose the following four non-mutually exclusive molecular mechanisms for channel deregulation in genetic RyR2 disease. The first mechanism involves altered cytosolic and/or luminal Ca^2+^ activation, which may account for both GoF (e.g., R4497C [70] and N4104K [72] increase cytosolic and luminal Ca^2+^ sensitivity, respectively) and LoF mutations (e.g., Q3774L [66] and A4860G [78] decrease cytosolic and luminal Ca^2+^ sensitivity, respectively). The second mechanism involves altered RyR2 intra- and inter-subunit interactions. For example, if the domain X interacts with domain Y to stabilize the closed channel, a mutation that disrupts the inter-domain X-Y interaction will be GoF (e.g., R2474S [20], R176Q [18]), whereas a mutation that enhances it will be LoF. On the other hand, if domain X interacts with domain Y to stabilize the open channel, a mutation that disrupts the inter-domain X-Y interaction will be LoF, whereas a mutation that enhances it will be GoF (e.g., R420Q [77]). The third mechanism involves altered interactions with accessory proteins (e.g., S2246L reducing FKBP12.6 binding [73]) as well as phosphorylation by protein kinases, which in turn will affect the second or third mechanisms. Finally, the fourth mechanism involves altered Ca^2+^ conductance and/or permeability, which may account for LoF mutations within the pore-forming S6 helix and the selectivity filter.

## 6. RyR2 Subcellular Geometry Influences Cardiac Ca^2+^ Release

### 6.1. RyR2 Clusters and Functional Ca^2+^ Release Units (CRU)

Early investigations using electron microscopy established that square-shape RyR2 tetramers (~30 × 30 nm) organize as discrete clusters in a quatrefoil crystalline arrangement on junctional SR membrane [84]. Purified RyR2-FKBP12.6 complexes can self-assemble in vitro into oblique multi-channel structures that appear to involve SPRY1 and FKBP12.6 interactions between neighboring clamp regions [85]. Recent advances in super-resolution optical microscopy have further enabled the detection and characterization of RYR2 organization at near single molecule level in both 2D [86,87] and 3D [88], which was previously not possible due to diffraction limitations of conventional confocal microscopes. These recent investigations in rodent, sheep, and human myocytes have shown that RyR2 clusters typically consist of ~15–20 channels that can be sparsely packed [86,88,89,90]. This is much smaller than previous estimates of ~100 RyR2 per cluster, calculated from thin sectioning that relied on assumptions of complete filling and circular geometry [91]. An emergent property of these clusters is the apparent allosteric regulation between channels, through some form of coupled gating or cross-talk, that enhances the fidelity of Ca^2+^ spark activation when [Ca^2+^] is increased, e.g., evoked by an action potential, but suppresses RyR2 opening at diastolic [Ca^2+^] [92]. Ca^2+^ spark amplitude was found to be fairly insensitive to cluster sizes greater than ~20 RyR2 (quantified indirectly using fluorescently-labeled FKBP12.6) [93]. Although larger clusters should increase the peak Ca^2+^ flux rate in proportion to the number of open channels times the trans-SR [Ca^2+^] gradient, this will more rapidly deplete junctional SR [Ca^2+^], thus reducing the driving force for Ca^2+^ release and cause a gradual loss of regenerative capacity and finally Ca^2+^ spark termination [94].

Ca^2+^ concentration decreases rapidly as it diffuses further from its source, therefore the probability of one cluster causing Ca^2+^-induced Ca^2+^ release (CICR) in distant clusters falls sharply with distance, and this limits uncontrolled activation. However, if two or more clusters are sufficiently close then the increase in cytosolic [Ca^2+^] can be enough to activate its neighbor, such that closely spaced RyR2 clusters (~100–150 nm edge-to-edge) are predicted to act together as functional Ca^2+^ release units (CRU) in the formation of Ca^2+^ sparks [86,92]. Ca^2+^ spark frequency showed a strong positive relationship with cluster size, indicating that larger clusters are more likely to initiate Ca^2+^ sparks [93] and support the notion that multiple clusters are involved in producing Ca^2+^ sparks.

### 6.2. Pathological Fragmentation of CRU

The break-up and fragmentation of CRU into smaller, dislocated clusters could contribute to the slower Ca^2+^ release kinetics and increased Ca^2+^ leak in inherited or acquired heart disease. Using stimulated emission-depletion (STED) microscopy, Macquaide et al. found that in a sheep model of chronic atrial fibrillation (AF) the mean distance between cluster centroids was shorter, without a significant change in the number of RyR2 per cluster, compared to healthy controls [89]. Consequently, more clusters fell within the CRU boundary (defined here as ≤150 nm) which increased the mean CRU area, but at a lower overall RyR2 density, suggesting greater fragmentation of CRU in this sheep model of AF. Reduced RyR2 density was also found in CRU from rats with right ventricular failure, using expansion microscopy combined with Airyscan super-resolution imaging [95]. Sheard et al. used geometrically realistic experimental maps of CRU to model the effect of cluster morphology on Ca^2+^ release and found the disrupted CRU often failed to fully activate within the typical time-course of a Ca^2+^ spark (~20 ms) [95]. The mechanism for CRU breakup remains to be established, but could include loss of accessory proteins tethering RyR2 clusters together, or simply CRU failing to keep up with cellular remodeling (e.g., hypertrophy in the case of heart failure and possibly RyR2 mutations [65]). Yin et al. used RyR2 immunolabeling and STED microscopy to investigate cluster morphology in mice with the CPVT-causing RyR2-R420Q^+/−^ missense mutation. RyR2 clusters were smaller on average in knock-in than WT controls, despite total RyR2 protein being similar between groups [77]. Junctophilin-2 may play a role in determining RyR2 clustering because overexpression increased the size of clusters [96]. Yin et al. found a reduced association between junctophilin-2 and RyR2 in RyR2-R420Q^+/−^ knock-in mice measured by co-immunoprecipitation, suggesting this could contribute to the redistribution of cluster size that results from this mutation [77]. In contrast to work in the experimental sheep heart failure [89], Munro et al. found RyR2 cluster and CRU properties to be unaltered in atrial myocytes from human patients with persistent AF versus patients without AF [90]. Here, the non-AF samples came from patients requiring coronary artery bypass graft surgery and may have experienced some degree of prior cardiac remodeling, which could contribute to the lack of CRU disruption observed. Alternatively, the sheep AF model may not fully recapitulate human atrial disease.

## 7. Arrhythmogenic Consequences of RyR2 Dysfunction

### 7.1. Ca^2+^ Sparks

Ca^2+^ sparks are brief events lasting ~30 ms that increase cytosolic Ca^2+^ by ~100 nM and have a spatial spread of ~2 µm full width at half maximum [4]. Unlike action potential evoked Ca^2+^ release, spontaneous Ca^2+^ sparks do not require Ca^2+^ entry through L-type Ca^2+^ channels to occur, but instead occur stochastically, usually at a low frequency between beats, due to the small but finite open probability of RyR2. RyR2 opening causes a localized increase in Ca^2+^ in the immediate cluster vicinity [97] that initiates strong positive CICR feedback, causing further channels in the CRU to open. The prevailing control theories of how Ca^2+^ sparks terminate all include some form of local depletion of junctional SR Ca^2+,^ as observed experimentally [98], which diminishes CICR from Ca^2+^ released from the SR into the dyad [94,99].

The remainder of this mini-review focuses on how changes in RyR2 behavior, associated with specific mutations, RyR2 modulators, and beta-adrenergic signaling, produce diverse phenotypes and potentially fatal cardiac arrhythmias with involvement from Ca^2+^ sparks and Ca^2+^ waves. The ability of influx and efflux pathways in the heart to “auto-regulate” Ca^2+^ levels is essential for understanding how these effects arise (see [100] for a detailed description). In brief, a change in the balance of influx and efflux pathways that favors net Ca^2+^ entry increases Ca^2+^ loading, whereas an increase in net Ca^2+^ efflux will cause unloading. This occurs because the steep load dependence of SR Ca^2+^ release [101] means that a transient reduction in SR load will cause: (1) a smaller Ca^2+^ transient amplitude, (2) less Ca^2+^-dependent inactivation of I_Ca_ (allowing more Ca^2+^ entry), (3) decreased activation of NCX (less Ca^2+^ removal), (4) reduced diastolic Ca^2+^ release, and (5) SR Ca^2+^ load will tend towards its previous state (if SR load is transiently increased then points 1–4 can be flipped).

As described in Section 5, the majority of pathological RyR2 mutations result in GoF phenotype, that manifests in cardiac myocytes as increased trigger ability of Ca^2+^ sparks to cytosolic or luminal Ca^2+^, from a low resting level (~1/s/100 µm in normal mouse ventricular myocytes) to much greater levels with pathological consequences. GoF RyR2 mutations and RyR2 sensitizing agents, such as caffeine, promote Ca^2+^ sparks, Ca^2+^ waves, and non-spark mediated leak, and, consequently, SR Ca^2+^ load is depressed during basal conditions (e.g., RyR2-R4496C [102], RyR2-R420Q [77]). Lower resting SR load probably explains why CPVT patients with GoF mutations typically have normal resting ECG [103] because the reduced SR load will somewhat limit spontaneous Ca^2+^ release. Arrhythmias typically manifest during periods of increased sympathetic activity, which increases Ca^2+^ entry and re-uptake pathways and shifts the SR towards greater operating Ca^2+^ levels that greatly promotes spontaneous Ca^2+^ release [102]. Ca^2+^ release refractoriness, measured using the ratio of the second:first Ca^2+^ spark occurring from the same CRU using a chemical modification of RyR2 with low concentration ryanodine [104], was reduced in a transgenic mouse model of the GoF CPVT-causing RyR2-R420Q^+/−^ mutation under both basal and isoproterenol stimulation, indicating increased CICR sensitivity [105]. Reduced refractoriness was also observed in the same RyR2 CPVT model using two-photon photolysis of a caged Ca^2+^ chelator [106] and in other models of GoF CPVT due to missense mutations (CSQ2-R33Q [107]) or knockout of CSQ2 [108].

### 7.2. Spontaneous Ca^2+^ Waves

An increase in Ca^2+^ spark frequency increases the likelihood that a spontaneous regenerative process of Ca^2+^-induced Ca^2+^ release, called a Ca^2+^ wave, will occur. Ca^2+^ waves propagate by the saltatory triggering of neighboring CRU and firing of Ca^2+^ sparks. For Ca^2+^ waves to propagate it is necessary that cytosolic [Ca^2+^] at nearby clusters increases sufficiently to activate nearby CRU spaced ~0.5 µm apart in the same Z-line [92], or across the much larger inter-Z line spacing (~1.8–2.0 µm). An increase in cytosolic or SR Ca^2+^ increases the interactions between CRU and the occurrence of propagating Ca^2+^ waves [109]. The likelihood of Ca^2+^ wave generation is predicted to be highly dependent on Z-disk spacing, by influencing the initial Ca^2+^ flux needed to raise Ca^2+^ above the threshold for CRU activation at nearby sites [110]. Shorter inter-sarcomere spacing has been reported in isolated ventricular myocytes from left ventricular heart failure in the spontaneously hypertensive rat [111] and right ventricular failure due to pulmonary arterial hypertension [112]. The shorter sarcomere spacing in right ventricular failure was partially reversed by chronic in vivo treatment with the β_1_-adrenergic receptor blocker, metoprolol [113], and this was proposed to contribute to the observed reduction in Ca^2+^ wave frequency and triggered activity, despite similar Ca^2+^ spark frequency between treated and untreated cells.

‘Rogue’ RyR2 located on the network SR outside of SR junctions could, in principle, facilitate Ca^2+^ wave propagation by reducing the longitudinal inter-CRU distance. Rogue RyR2 occur infrequently in normal ventricular myocytes [86] and Ca^2+^ sparks occur almost exclusively from RyR2 clusters in Z-lines [114], suggesting they may have a minor role in Ca^2+^ wave propagation in normal cardiomyocytes. There was no difference in the proportion of non-Z-line RyR2 in human persistent AF [90]; however, the proportion of RyR2 clusters between Z-lines was greater in atrial myocytes from sheep with chronic AF [89], so it is feasible that rogue RyR2 could contribute to increased spontaneous Ca^2+^ waves in a pathological context. Rogue RyR2 could be involved in substantial non-Ca^2+^ spark mediated leak, involving brief openings of small size clusters that would minimally raise cytosolic Ca^2+^ and would be below the optical limit of detection using fluorescent Ca^2+^ sensitive dyes [115]. Increased Ca^2+^ spark-independent leak in failing rabbit ventricular myocytes contributed to reduced SR Ca^2+^ load [101] and would be expected to impair contractility due to the steep dependence of evoked Ca^2+^ release on SR Ca^2+^ load during normal excitation–contraction coupling.

### 7.3. Delayed Afterdepolarizations (DADs)

The Na^+^/Ca^2+^ exchanger is one of the main Ca^2+^ removal pathways that restore cytosolic Ca^2+^ during diastolic Ca^2+^ waves back towards resting levels. Near resting membrane potentials, NCX works in the forward mode to remove 1 Ca^2+^ ion from the cytosol in return for 3 Na^+^ ions entering. This net electrogenic action produces a transient inward current that depolarizes the resting membrane potential, causing events called delayed afterdepolarizations (DADs). If DADs reach the threshold for Na^+^ channel activation, then triggered action potentials can occur and activate surrounding cells resulting in ectopic electrical activity in the heart that can degenerate into ventricular tachycardias that can lead to sudden cardiac death. DADs are believed to be the underlying event responsible for arrhythmia in CPVT [116]. Fortunately, there is a large mismatch between the current generated by DADs in a single cell (the source) that is shared between all the surrounding electrically-connected cells (the sink), and therefore occasional Ca^2+^ waves in cells in the heart are unlikely to propagate the electrical activity further. Using optogenetic activation of channelrhodopsin-2 expressing cardiac-specific transgenic mice, it was estimated that ~2000 myocytes must depolarize near-simultaneously to initiate premature ventricular contractions [117]. The stochastic nature of DADs makes the probability of this happening by chance highly unlikely. However, this probability increases if Ca^2+^ regulation is deranged by factors causing RyR2 GoF phenotype and if there is coupling to entrain oscillations between cells. Indeed, Borile et al. found that during beta-adrenergic stimulation in ventricular heart slices from mice harboring the CPVT-causing RyR2-R2474S heterozygous missense mutation, cells surrounding those that exhibited DADs also had a greater probability of Ca^2+^ waves occurring almost synchronously [118].

### 7.4. Early Afterdepolarizations (EADs)

Early afterdepolarizations (EADs) occur during repolarization due to a mismatch between outward K^+^ currents and inward currents (such as I_Na_, I_Ca_, I_NCX_) and are believed to be a major source of ventricular tachyarrhythmias [119]. Similar to DADs, spontaneous Ca^2+^ release during repolarization increases inward I_NCX_ current and has been recognized as being involved in EADs for many years [120] (and see [121] for a recent review). I_NCX_ is a non-regenerative current (current decreases as membrane potential increases) and was believed to contribute mainly by delaying repolarization and allowing time for reactivation of other inward currents (such as I_Na_ or I_Ca_) to drive the upstroke of the EAD [120,122,123]. Stochastic SR Ca^2+^ release, in the form of ‘late Ca^2+^ sparks’ [124], occurs during the normal Ca^2+^ transient decay triggered by I_Ca_ during the AP plateau and increased cytosolic Ca^2+^. The frequency of late Ca^2+^ sparks was greater in failing rabbit ventricular myocytes [125,126] and could initiate propagating wave-like ‘Ca^2+^ ripples’. Under conditions of increased Ca^2+^ load, late Ca^2+^ sparks-mediated I_NCX_ could take a dominant role in driving the depolarization [125]. This conclusion is supported by computer modeling by Zhong et al. demonstrating that increased late Ca^2+^ release due to hyperactive RyR2 (increased Ca^2+^ sensitivity and reduced refractoriness) contributes to EAD genesis by increasing I_NCX_ [127]. Late Ca^2+^ sparks can fire more than once during a Ca^2+^ transient, after an apparent refractory period that is likely limited by the rate of junctional Ca^2+^ refilling [104,124], providing a mechanism to sustain or increase I_NCX_ at positive membrane potentials. The involvement of SR Ca^2+^ release in EADs is further apparent in the case of conditional SERCA2a knockout mouse ventricular myocytes [128]. The SR Ca^2+^ store is essentially absent in knockout mice, demonstrated by the absence of Ca^2+^ sparks, meaning contraction is solely maintained by an increase in trans-sarcolemmal Ca^2+^ entry through I_Ca_ and a slight decrease in K^+^ current density. Despite such major electrophysiological adaptations, myocytes from SERCA2a knockout mice do not exhibit greater EAD incidence than control mice. It is worth noting that the major ion channels implicated in causing EADs are highly coupled to each other (Figure 3), and that knocking-out or inhibiting one part of the pathway will affect the others and tend to exaggerate its (patho)physiological role. Alternatively, combining experimental findings with mathematical modeling is a powerful way to dissect the bidirectional coupling between Ca^2+^ ↔ membrane potential that is difficult to interpret from experiments alone [123,125].

Factors that increase RyR2 Ca^2+^ sensitivity, or decrease refractoriness, are likely to increase the frequency of late Ca^2+^ sparks. Low dose caffeine caused a biphasic Ca^2+^ transient decay in rat ventricular myocytes which, although not directly measured using confocal microscopy, would be consistent with RyR2 sensitization promoting cytosolic Ca^2+^ triggered late Ca^2+^ sparks [129]. Post-translational modification of RyR2 by PKA and/or CaMKII phosphorylation may increase the RyR2 Ca^2+^ sensitivity and has been attributed as a possible cause of the hyperactive RyR2 observed in heart failure, although the physiological consequences and precise locations of phosphorylation are still the matter of ongoing investigation (for a recent review see [130]). Guo et al. investigated the effect of overexpression of CaMKII𝛿c on RyR2 Ca^2+^ release properties in transgenic mice on a phospholamban knockout background to avoid the confounding influence of altered SR Ca^2+^ load (because CaMKII can also phosphorylate phospholamban and stimulate SERCA2a Ca^2+^ uptake) [131]. They found that diastolic Ca^2+^ sparks had a faster rate of rising and shorter refractory period between consecutive Ca^2+^ sparks, possibly suggesting more RyR2 channels opening during each event due to increased Ca^2+^ sensitivity. The Ca^2+^ transient decay was slower and more late Ca^2+^ sparks occurred in CaMKII𝛿c over-expressing mice than phospholamban knockout controls, which could be caused by RyR2 Ca^2+^ hypersensitivity, but could also be due to CaMKII𝛿c facilitation of I_Ca_ [132] that would also prolong the Ca^2+^ transient and promote late Ca^2+^ sparks [124] Liu et al. found abnormal Ca^2+^ release in ventricular myocytes from mice with the CPVT-causing CSQ2-R33Q mutation that results in a GoF RyR2 phenotype, suggesting systolic Ca^2+^ handling could also be affected by GoF mutations [133].

The rare LoF RyR2 mutations described in Table 2 are associated with atypical electrical behavior including EADs. Zhao et al. investigated the properties of excitation-contraction coupling in transgenic mice with the LoF RyR2-A4860G^+/−^ missense mutation [134]. Under basal conditions, the amplitude and rate of rising of the Ca^2+^ transient were depressed and SR Ca^2+^ load was greater in RyR2-A4860G^+/−^ ventricular myocytes compared to WT cells. Continued Ca^2+^ entry during the action potential via I_Ca_ with less efficient evoked SR Ca^2+^ release was proposed to increase net Ca^2+^ entry, overloading the SR, and resulting in frequent prolonged systolic Ca^2+^ release and EADs under basal conditions, and at greater frequency during stimulation with isoproterenol. They proposed that the extended Ca^2+^ release during EADs help unload the SR in this LoF model, although this will be balanced somewhat by the increased Ca^2+^ entry through I_Ca_ at depolarized membrane potentials. Alternatively, it was recently proposed that the reduced Ca^2+^ transient rate of rise in RyR2-A4860G^+/−^ could cause less Ca^2+^-dependent inactivation and therefore prolong I_Ca_ during the action potential plateau, which would increase action potential duration both directly and indirectly (through increased I_NCX_), although this remains to be demonstrated experimentally [135]. Sun et al. found that transgenic mice carrying the LoF RyR2-D4646A^+/−^ mutation also showed increased EADs and SR Ca^2+^ load in isolated myocytes and re-entrant arrhythmias following burst pacing in isolated hearts, although this occurred without a change in steady state Ca^2+^ transient amplitude or rate of rise, and a markedly increased I_Ca_ density [66]. These initial findings suggest the principal arrhythmogenic substrate may be different in LoF compared to GoF RyR2 disorders (although there is likely some cross-over in mechanisms) (Figure 3).

## 8. Conclusions

The list of mutations in genes encoding RyR2 and its regulatory proteins associated with pro-arrhythmic phenotype continues to grow. These are revealing new insights into the structure/function relationship of RyR2 that is communicated both within and between subdomain interactions of this critical ion channel. It is now generally accepted that there is no common molecular defect, no single unifying hypothesis to explain how RyR2 GoF mutations result in typical CPVT. Moreover, molecular characterization of site-specific mutations may help explain the apparent diverging phenotypes in GoF and LoF RyR2 mutations. However, as LoF mutations have only fairly recently been the subject of attention, further work is needed to establish the extent to which their arrhythmogenic substrates are unique, or whether there is also some overlap with GoF mutations, as this may have important implications for the optimal therapeutic approach. Improved capabilities to investigate subcellular changes in RyR2 behavior and Ca^2+^ release properties in the native environment using advanced imaging technologies offers an exciting future direction that promises to better our understanding of how cardiac excitation-contraction coupling is regulated in health and disease.

## Figures and Tables

**Figure 1 biomolecules-12-01030-f001:**
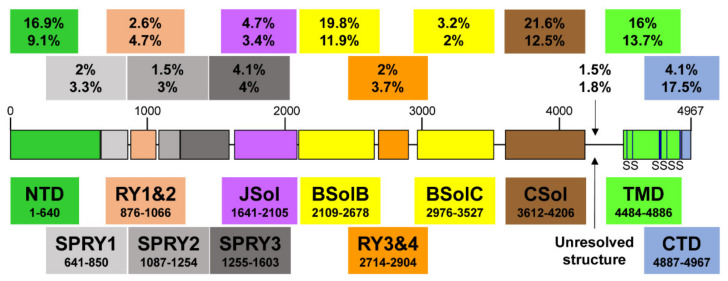
A schematic illustration of human RyR2 domain organization and boundaries. Color-coded distribution of mutations associated with disease (top value) and their frequency within individual structural domains (bottom value). The six transmembrane segments (S) are represented by a vertical line. See Figure 2 below for a corresponding cartoon representation showing the arrangement of domains in 3D (using same color coding).

**Figure 2 biomolecules-12-01030-f002:**
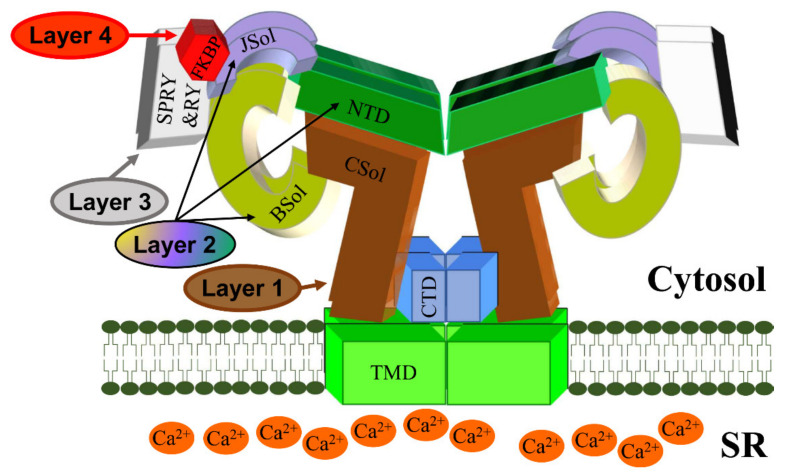
A schematic representation of the RyR2 3D architecture depicting 4 regulatory layers from the cytoplasmic side. The pore-forming region, comprised of the TMD and CTD, is modulated by the CSol (Layer 1), NTD, JSol and BSol (Layer 2), SPRY1/2/3 and RY1/2/3/4 (Layer 3), and accessory proteins (Layer 4), e.g., FKBP. The colors of regions in the RyR2 3D structure correspond to the colors in the RyR2 sequence in Figure 1.

**Figure 3 biomolecules-12-01030-f003:**
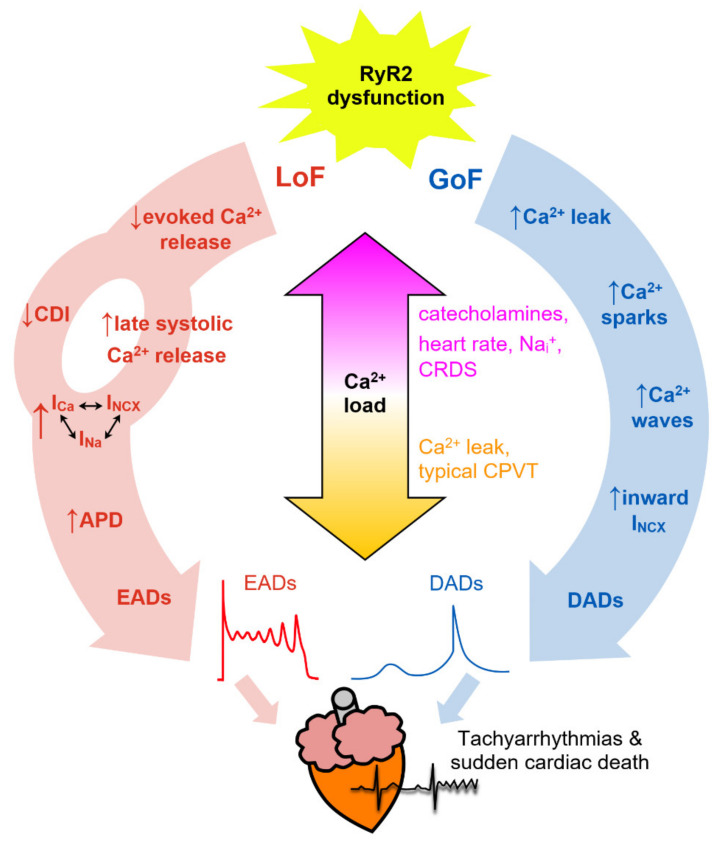
Simplified schematic to illustrate the proposed pathways linking RyR2 loss-of-function (LoF, red) and gain-of-function (GoF, blue) abnormalities to pro-arrhythmic behavior. In GoF, excess Ca^2+^ leak leads to diastolic Ca^2+^ waves and DADs which tend to decrease the Ca^2+^ load. In LoF, evoked Ca^2+^ release is impaired resulting in Ca^2+^ loading and less Ca^2+^-dependent inactivation (CDI) of I_Ca_, ultimately resulting in prolonged systolic Ca^2+^ release and EADs. Increasing Ca^2+^ load worsens Ca^2+^ regulation and hastens arrhythmia development.

**Table 1 biomolecules-12-01030-t001:** RyR2 domain distribution of disease-linked mutations.

Domain	Coordinates (Amino Acids)	Length (Amino Acids)	Number of Mutations	Mutation Frequency Overall (%)	Mutation Frequency within Domain (%)
NTD	1–640	640	58	16.9	9.1
SPRY1	641–850	209	7	2.0	3.3
RY1&2	876–1066	190	9	2.6	4.7
SPRY2	1087–1254	167	5	1.5	3.0
SPRY3	1255–1603	348	14	4.1	4.0
JSol	1641–2105	464	16	4.7	3.4
BSolB	2109–2678	569	68	19.8	11.9
RY3&4	2714–2904	190	7	2.0	3.7
BSolC	2976–3527	551	11	3.2	2.0
CSol	3612–4206	594	74	21.6	12.5
Unresolved	4207–4483	276	5	1.5	1.8
TMD	4484–4886	402	55	16.0	13.7
CTD	4887–4967	80	14	4.1	17.5

**Table 2 biomolecules-12-01030-t002:** RyR2 loss-of-function mutations.

Mutation	Domain	Disease	References
Duplication ^1^	-	CRDS	[67]
L433P ^2^	NTD	ARVD2	[22]
G570D	NTD	CRDS	[79]
Q2275H	BSolB	CRDS	[80]
Q3774L	CSol	CRDS	[66]
Q3925E	CSol	CRDS	[79]
I3995V	CSol	CRDS	[66]
M4109R	CSol	CRDS	[79]
D4112N	CSol	CRDS	[66]
A4142T	CSol	CRDS	[81]
E4146D	CSol	long QT syndrome	[82]
E4146K	CSol	CRDS	[83]
R4147K	CSol	CRDS	[79]
S4168P	CSol	long QT syndrome	[82]
T4196I	CSol	CRDS	[66]
A4203V	CSol	CRDS	[79]
A4204V	CSol	CRDS	[79]
E4451del	Unresolved	CRDS	[80]
F4499C	TMD	CRDS	[80]
K4594Q	TMD	long QT syndrome	[82]
V4606E	TMD	CRDS	[80]
R4608Q	TMD	CRDS	[80]
R4608W	TMD	CRDS	[80]
D4646A	TMD	CRDS	[66]
I4855M	TMD	left ventricular noncompaction	[62]
A4860G ^3^	TMD	CRDS	[66,78]
Q4879H	TMD	CRDS	[66]
G4935R	TMD	CRDS	[83]
S4938F	TMD	torsade de pointes	[59]

Notes: ^1^ Homozygous duplication of the promoter region and exons 1–4. ^2^ L433P has both GoF and LoF characteristics. ^3^ Idiopathic ventricular fibrillation that was reclassified as CRDS.

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
