# Peer review of "Molecular, Subcellular, and Arrhythmogenic Mechanisms in Genetic RyR2 Disease"

_biomolecules, 2022, doi:10.3390/biom12081030_

Round 1
Reviewer 1 Report
This manuscript reviews the molecular, subcellular and arrhythmogenic mechanism in typical and atypic CPVT. It is very well and clearly written and concisely summarizes the latest findings of the mechanisms of the occurrence of arrythmia in gain of function and most interestingly in loss of function mutations in RYR2. The figures are clear and support the manuscript very well. This is clearly not a clinical review, but still in the short clinical section only the ESC Guidelines of 2015 are mentioned. These are still current, but already 7 years old and some new important clinical findings like a recent report on the superiority of nonselective beta blockers (in particular nandolol) over selective beta blockers should be mentioned (see PMID 35353122). I recommend this manuscript for publication with very minor revisions.
Specific points:
- Figure 1: Please specify the meaning of the “SS SSSS” in the Figure Legend.
- Tables: Please provide legends for the Tables if consistent with journal style.
Reviewer 2 Report
This mini-review article by Fowler et al, summarizes what's known about RyR structure and function as it relates to specific protein interactions and contributions to disease. The review is well written, but the first figure is redundant and unnecessary. Information on that figure is better represented in figure 2 and Table 1. Figure 2 would benefit from professional illustration service.
